# BAME: Block-Aware Mask Evolution for Efficient N:M Sparse Training

**Chenyi Yang** [1]  **Wenjie Nie** [1]  **Yuxin Zhang** [1]  **Yuhang Wu** [1]  **Xiawu Zheng** [1]  **Guannan Jiang** [2]  **Rongrong Ji** [1 3]

## Abstract

N:M sparsity stands as a progressively important tool for DNN compression, achieving practical speedups by stipulating at most N non-zero components within M sequential weights. Unfortunately, most existing works identify the N:M sparse mask through dense backward propagation to update all weights, which incurs exorbitant training costs. In this paper, we introduce BAME, a method that maintains consistent sparsity throughout the N:M sparse training process. BAME perpetually keeps both sparse forward and backward propagation, while iteratively performing weight pruning-and-regrowing within designated weight blocks to tailor the N:M mask. These blocks are selected through a joint assessment based on accumulated mask oscillation frequency and expected loss reduction of mask adaptation, thereby ensuring stable and efficient identification of the optimal N:M mask. Our empirical results substantiate the effectiveness of BAME, illustrating it performs comparably to or better than previous works that fully maintaining dense backward propagation during training. For instance, BAME attains a 72.0% top-1 accuracy while training a 1:16 sparse ResNet-50 on ImageNet, eclipsing SR-STE by 0.5%, despite achieving $2.37\times$ training FLOPs reduction. Code is released at https://github.com/BAME-xmu/BAME.

## 1. Introduction

In recent years, the vision community has precipitously bolstered the performance of Deep Neural Networks (DNNs) across various tasks, including image classification (He

et al., 2016), object detection (He et al., 2017a), and semantic segmentation (Girshick et al., 2014), *etc*. These progressions are chiefly driven by an augmented parameter burden and an increasingly onerous computational cost. Regrettably, this tendency presents significant impediments for the deployment of DNNs on resource-constrained edge devices like smartphones and various Internet of Things (IoT) apparatuses. Consequently, there has been a proliferation of interest in model compression research (Hubara et al., 2016; Howard et al., 2017; Lin et al., 2020), with the explicit objective of reducing the model's computation and parameter complexity whilst preserving comparable performance to the original model, thereby alleviating the deployment tribulations experienced with DNNs.

Among these techniques, network sparsity has proven many successes (Han et al., 2015; LeCun et al., 1989; Luo et al., 2017) by zeroing weights to yield lightweight, sparse networks at different granularity levels, from fine to coarse. Fine-grained sparsity (unstructured sparsity) (LeCun et al., 1989; Ding et al., 2019) removes individual weights and is demonstrated to well retain performance even at high sparsity rates. Regrettably, the deployment of such fine-grained sparse networks onto mainstream hardware systems becomes exceptionally challenging, given the irregular matrix patterns created by sparse weights. In contrast, coarse-grained sparsity, otherwise known as structured sparsity, (He et al., 2017b; Lin et al., 2020) procures substantial acceleration, purging whole convolution filters in the process (Liu et al., 2019; Lin et al., 2020). Nevertheless, structured sparsity can experience severe performance degradation, especially under high sparsity conditions. Recent developments indicate N:M sparsity as an auspicious avenue towards effectively balancing the dual requirements of acceleration and performance retention (Zhou et al., 2021; Pool & Yu, 2021). By imposing a restriction of, at most, N non-zero elements within M sequential weights throughout the input channel dimension, N:M sparsity can substantially enhance the performance of structured sparsity, while concurrently assuring swift inference, ably facilitated by the N:M sparse tensor core (Nvidia, 2020).

The crux of maintaining the performance of N:M sparse networks lies in identifying the optimal N:M sparsity mask. To achieve this, prevalent methodologies involve updating all weights during training to determine the most effective

---

*Equal contribution [1]Key Laboratory of Multimedia Trusted Perception and Efficient Computing, Ministry of Education of China, Xiamen University [2]Contemporary Amperex Technology Co., Limited (CATL) [3]Institute of Artificial Intelligence, Xiamen University. Correspondence to: Rongrong Ji <rrji@xmu.edu.cn>.

*Proceedings of the 42nd International Conference on Machine Learning*, Vancouver, Canada. PMLR 267, 2025. Copyright 2025 by the author(s).

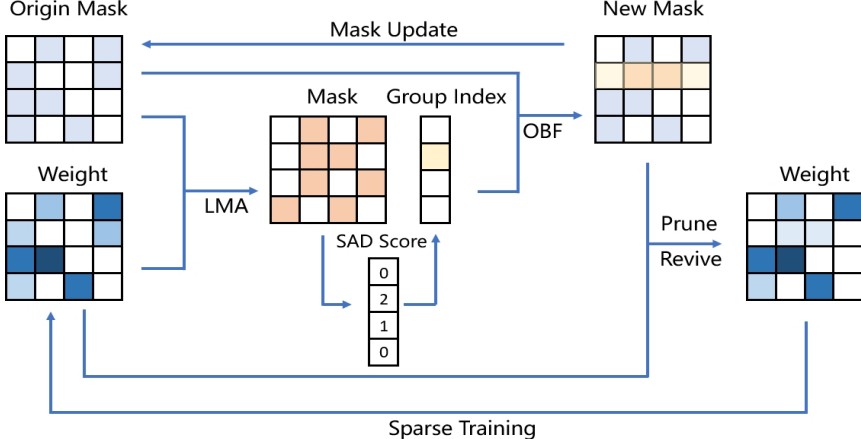

Figure 1: Framework of BAME. It iteratively performs weight pruning-and-regrowing through Loss-Aware Mask adaption (LMA) and Oscillation-aware Block Freezing (OBF), which leads to stable and efficient location for the optimal N:M mask.

N:M mask, adopting a straight-through estimator to approximate the gradients of the pruned weights (Zhou et al., 2021; Zhang et al., 2023b) or learning the importance criteria for all weights (Zhang et al., 2022). Despite their efficacy, the computation of dense gradients invariably imposes a substantial training overhead. Notably, the reduction of training costs has been a focal research point within the sparsity comunity in recent years (Liu et al., 2021; Evci et al., 2020; Dettmers & Zettlemoyer, 2019). With the ever-growing size of cutting-edge models, the significant computational demands and energy consumption of training sparse networks are escalating critical environmental, ethical, and financial concerns. Consequently, the development of efficient and scalable N:M sparse training methods is paramount, potentially even more urgent, to support the widespread accessibility and democratization of DNNs.

In this paper, we present BAME as a way of maintaining consistent sparsity in both forward and backward propagation throughout the N:M sparse training process. As shown in Figure 1, BAME escapes from dense weight's update through block-aware N:M mask evolution. It specifically executes weight pruning-and-regrowing within each consecutive M weights in order to adapt the sparse mask. Such mask evolution occurs solely when the detrimental effects on loss caused by pruning a certain weight is outweighed by the gain in loss from restoring another already pruned weight. Concurrently, we selectively adapt the mask of N:M blocks, as some blocks are experimentally observed to exhibit frequent oscillations on their masks during training, leading to unstables loss landscape. To this end, we employ exponential moving averaging (EMA) to accumulate the incidence of mask fluctuations for each block, choosing those with fewer fluctuations for mask evolution to ensure stable optimization for the N:M sparse network during training. In this manner, BAME can stably optimize the N:M mask while conducting N:M sparse training in a dense-backward-free efficient manner.

We conduct extensive experiments on validating the effectiveness and efficacy of BAME for N:M sparse training. The results show that BAME is able to get state-of-the-art performance when training N:M sparse networks across a wide range of sparse pattern, datasets, and prevailing DNNs, even with much fewer training FLOPs compared with existing work. Illustratively, BAME attains a 72.0% top-1 accuracy while training a 1:16 sparse ResNet-50 on ImageNet, eclipsing SR-STE (Zhou et al., 2021) by 0.5%, while using far less training FLOPs. Our work provides fresh insights in N:M sparse training without dense weight updates and we anticipate that BAME will not only equip practitioners with a robust training tool but also lay the groundwork for subsequent explorations into the training efficiency of N:M sparsity.

## 2. Related Work

### 2.1. Network Sparsity

By removing redundant weights to eliminate the parameter and FLOPs burden, network sparsity has emerged as a fervent area of research over the last decade (LeCun et al., 1989; Han et al., 2015; Louizos et al., 2017). Traditional approaches can broadly be classified into two categories based on their pruning granularity: unstructured and structured sparsity. The former involves the elimination of individual weights at any location within the network, achieving sparsity at a fine-grained level (Han et al., 2015; Lee et al., 2019; Ding et al., 2019). In essence, unstructured sparsity can rival the performance of their dense counterparts even at exceedingly high sparsity ratios, such as 90% (Mostafa & Wang, 2019). Nonetheless, the generated sparse weight tensors generally precludes acceleration on standard hardware platforms unless the sparsity ratio reaches or exceeds

95% (Wang). Conversely, structured sparsity achieves notable acceleration by extensively removing entire weight rows or convolution filters (Luo & Wu, 2020; Lin et al., 2020). Regrettably, structured sparsity often leads to substantial performance degradation at sparsity levels exceeding 50%, attributed to the constraints imposed on sparsity flexibility. Diverging from conventional sparsity granularities, this paper delves into N:M sparsity that removes weight in an mid-level granularity and has garnered significant research interest in recent years (Zhou et al., 2021; Sun et al., 2021; Pool & Yu, 2021).

## 2.2. N:M Sparsity

The recent development of N:M sparsity upholds the conservation of N-out-of-M consecutive weights in DNNs (Nvidia, 2020; Pool & Yu, 2021; Sun et al., 2021; Zhou et al., 2021; Chmiel et al., 2021; Hubara et al., 2016; Zhang et al., 2022). Supported by the NVIDIA Ampere Core (Ronny Krashinsky, 2020), N:M sparsity fosters superior storage and computational efficiency, establishing an immaculate harmony between model efficiency and precision, outdoing both unstructured and structured sparsity. To illustrate, 2:4 sparsity can realize 2× speedups on an NVIDIA A100 GPU, while unstructured sparsity might further decelerate the inference speed at identical levels of sparsity. As trailblazing work, ASP (Nvidia, 2020) employs a traditional tri-phase workflow encompassing model pre-training, high-magnitude weight extraction (Han et al., 2015), and network fine-tuning. (Zhou et al., 2021) subsequently proposed to train N:M sparse network from scratch by introducing the Sparse-refined Straight-Through Estimator (SR-STE). More specifically, N-out-of-M weights of higher magnitudes are selected in each forward pass, whileall weights are updated during the backward phase, utilizing the STE estimator, paired with a uniquely designed sparse penalty term. LBC (Zhang et al., 2022) further recasts N:M sparsity as a combinatorial problem, learning the optimal mask for each N:M block. MaxQ (Xiang et al., 2024) utilizes a multi-axis query to generate soft N:M masks during training to further improve the performance. Despite their effectiveness in preserving the performance of sparse networks, most existing works require dense backward propagation to update all weights to discover the optimal N:M mask, leading to massive training burden and memory cost. Our proposed BAME in this paper diverges from existing N:M methods as it performs both sparse forward and backward propagation during the entire training process, substantially alleviating the training cost.

## 2.3. Sparse Training

Sparse training, which dynamically adjusts the sparse masks throughout the training process, has recently emerged as a promising solution to enhance the training efficiency of network sparsity (Hoefler et al., 2021; Evci et al., 2020; Han et al., 2015; Liu et al., 2021). The most representative method RigL (Evci et al., 2020) prunes weights of smaller magnitudes during inference and subsequently regrows the same quantity of weights based on their gradient values throughout backward propagation. Sparse Momentum (Dettmers & Zettlemoyer, 2019) employs the mean momentum magnitude of each layer as a benchmark for redistributing parameters. (Kusupati et al., 2020) proffer layer-wise learnable thresholds strategizing the reallocation of parameters across layers. Moreover, (Liu et al., 2021) proposed to gradually increase the sparsity level during training to further enhance the performance of sparse networks. While these approaches predominantly concentrate on boosting unstructured sparsity, our endeavor in this paper differs by targeting the training of N:M sparse networks, innovatively designing a block-aware selection mechanism for pruning and reviving N:M sparse weights.

## 3. Methodology

### 3.1. Background

We first recap basic preliminaries of N:M sparsity. For simplicity, we take the weights from a specific layer within DNNs for illustration. N:M sparsity forces at most N out of M consecutive weights in the weight row to have non-zero values. The weights can be therefore grouped into $K$ blocks where each block contains M consecutive weights, denoted as $\mathbf{W} \in \mathbb{R}^{K \times M}$. And then, N:M sparsity can be formulated as multiplying $\mathbf{W}$ with a binary mask $\mathbf{B} \in \mathbb{R}^{K \times M}$, with the following objectives:

$$\min_{\mathbf{W}, \mathbf{B}} \mathcal{L}(\mathbf{W} \odot \mathbf{B}; \mathcal{D}) \quad s.t. \quad \|\mathbf{B}_{k,:}\|_0 = \mathrm{N}, \qquad (1)$$

where $k = 1, 2, ..., K$, $\odot$ is the point-wise element-wise multiplication, $\mathcal{L}(\cdot)$ denotes training loss function and $\mathcal{D}$ represents the observed training dataset, respectively. The zero elements in $\mathbf{B}$ indicate the removal of corresponding weights in the network, and vice versa.

**Challenge of N:M sparse training.** The crux of optimizing Equation (1) falls into locating high-quality masks that correctly preserve important weights. As a pioneer work, ASP (Nvidia, 2020) chooses to mask out weights that have lower magnitudes, intuitively reducing the output derivation between dense pre-trained weights and N:M sparse weights. Nevertheless, the pre-training phase unavoidably carries huge training burden. In the literature, a more popular way to obtain the sparse mask is performing training-time weight selection by updating all weights (Zhou et al., 2021; Zhang et al., 2022; Fang et al., 2022; Zhang et al., 2023b). Particularly, the straight-through-estimator (STE) (Bengio et al., 2013) is leveraged to calculate the gradient of all weights, since the currently removed weights always receive no gradient as their corresponding multiplied masks are 0s. Formally,

the gradients of $\mathbf{W}$ are derived as

$$\frac{\partial \mathcal{L}}{\partial \mathbf{W}} = \frac{\partial \mathcal{L}}{\partial (\mathbf{W} \odot \mathbf{B})} \odot \mathbf{B} \approx \frac{\partial \mathcal{L}}{\partial (\mathbf{W} \odot \mathbf{B})} \odot \mathbf{1}. \quad (2)$$

In this vein, all weights can be updated during the training process. By dynamically selecting weights with higher magnitude, such N:M sparse training can effectively boost the model performance, even without reliance on pre-trained weights. Despite recent efforts to further enhance N:M sparse training through additional norm constraints on pruned weights (Zhou et al., 2021) or gradual sparsity (Fang et al., 2022), one significant concern remains that dense back-propagation and weight updates continue to incur substantial resource consumption, posing challenges to scenarios with limited resources.

In this paper, we address the above hindrance of training inefficiency by proposing Block-Aware Mask Evolution (BAME), a method that ensures consistent sparsity throughout the forward and backward propagation phases of the N:M sparse training process. The unique contribution of BAME encompasses loss-aware mask adaption (LMA) that prune-and-revive weights to effectively decrease the training loss, and oscillation-aware block selection (OBS), limiting mask modifications within blocks demonstrating high-frequency mask oscillations, thus stabilizing the N:M training process. We introduce these two components as follows.

### 3.2. Loss-aware Mask Adaption

Owing to the great benefit of training cost reduction, adapting the sparse mask during training while escaping from dense gradient calculation has been a hot topic within traditional unstructured sparsity literature (Evci et al., 2020; Dettmers & Zettlemoyer, 2019; Liu et al., 2021; Jayakumar et al., 2020). The central philosophy of these methods involves performing a global pruning and revival based on instantaneous gradient information every few training iterations. Specifically, several of the weights with the highest gradients among all pruned weights are restored and the same number of weights with the lowest magnitude among all retained weights are pruned, therefore reducing the loss to the fastest extent.

Regrettably, prior methodologies for globally altering the sparse topology are unsuitable within the context of N:M sparsity. Following a fixed sparsity budget for each N:M block, pruning-and-reviving of weights can only be carried out in each independent N:M block. This presents substantial risks for the mask adaptation: The gradients of the weights in the same block are likely to have minor differences due to the continuous input received, as is the magnitude of the weights. Hence, directly applying traditional sparse methods can have high possibility of resulting in the recovery of weights yielding less loss benefit compared to the disruption caused by weight pruning, even if

the pruned weights have the smallest magnitude within the N:M block.

To address this challenge, we introduce loss-aware mask adaption (LMA) that ensures weight pruning-and-reviving always lead to loss decrease during training. LMA performs static sparse training in both forward and backward propagation, while only calculating dense gradient to perform mask adaption every $\Delta T$ iteration. Here we use a specific N:M block $\mathbf{W}_k \in \mathbb{R}^M$ to illustrate the mask adaption procedure. Considering a currently preserved weight $\mathbf{W}_{k,i}$ where $\mathbf{B}_{k,i} = 1$, the loss change, denoted as $\Delta \mathcal{L}(\mathbf{W}_{k,i})$, upon its removal can be approximately derived using first-order Taylor expansion (Molchanov et al., 2017) as:

$$\Delta \mathcal{L}(\mathbf{W}_{k,i})$$
$$= |\mathcal{L}(\mathbf{W} \odot \mathbf{B};\ \mathcal{D},\ \mathbf{B}_{k,i} = 0) - \mathcal{L}(\mathbf{W} \odot \mathbf{B};\ \mathcal{D},\ \mathbf{B}_{k,i} = 1)|$$
$$\approx |\mathcal{L}(\mathbf{W} \odot \mathbf{B};\ \mathcal{D},\ \mathbf{B}_{k,i} = 1) - \frac{\partial \mathcal{L}}{\partial (\mathbf{W} \odot \mathbf{B})_{k,i}}(\mathbf{W}_{k,i} - 0)$$
$$+ R_1(B_{k,i} = 0) - \mathcal{L}(\mathbf{W} \odot \mathbf{B};\ \mathcal{D},\ \mathbf{B}_{k,i} = 1)|.$$
$$(3)$$

If we ignore the first-order remainder $R_1(\mathbf{B}_{k,i} = 0)$, then:

$$\Delta \mathcal{L}(\mathbf{W}_{k,i}) \approx \left| \frac{\partial \mathcal{L}}{\partial (\mathbf{W} \odot \mathbf{B})_{k,i}} \mathbf{W}_{k,i} \right|. \quad (4)$$

Similarly, if we consider reviving a currently removed weight $\mathbf{W}_{k,j}$ back, the loss change $\Delta \mathcal{L}(\mathbf{W}_{k,j}) = 0$ can be derived as:

$$\Delta \mathcal{L}(\mathbf{W}_{k,j})$$
$$= |\mathcal{L}(\mathbf{W} \odot \mathbf{B};\ \mathcal{D},\ \mathbf{B}_{k,j} = 1) - \mathcal{L}(\mathbf{W} \odot \mathbf{B};\ \mathcal{D},\ \mathbf{B}_{k,j} = 0)|$$
$$\approx |\mathcal{L}(\mathbf{W} \odot \mathbf{B};\ \mathcal{D},\ \mathbf{B}_{k,i} = 0)$$
$$- \frac{\partial \mathcal{L}}{\partial (\mathbf{W} \odot \mathbf{B})_{k,j}} \left( 0 - (0 - \eta \frac{\partial \mathcal{L}}{\partial (\mathbf{W} \odot \mathbf{B})_{k,j}}) \right)$$
$$+ R_1(B_{k,i} = 1) - \mathcal{L}(\mathbf{W} \odot \mathbf{B};\ \mathcal{D},\ \mathbf{B}_{k,i} = 0)|$$
$$\approx \eta \left( \frac{\partial \mathcal{L}}{\partial (\mathbf{W} \odot \mathbf{B})_{k,j}} \right)^2,$$
$$(5)$$

where $\eta$ is the current learning rate. Based on the preceding derivation, we can articulate the following conclusions. On one hand, Eq. (4) tells that for the preserved weights, pruning those with comparably minor $\Delta \mathcal{L}(\mathbf{W}_{k,i})$ ensures the loss does not undergo notable alterations. This perspective concurs with traditional network sparsity knowledge (Molchanov et al., 2017; Zhang et al., 2023a). Conversely, considering the presently pruned weights, their revival will invariably benefit the minimization of loss as observed in the derivation of Eq. (5). Simultaneously, it bears mentioning that restoring weights with significantly larger $\Delta \mathcal{L}(\mathbf{W}_{k,i})$ will induce the most substantial degree of loss mitigation. Therefore, at each mask adaption cycle,

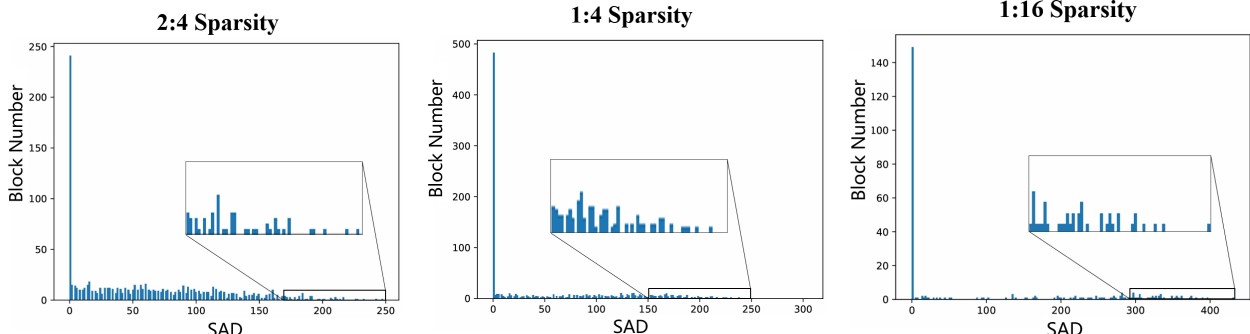

Figure 2: Sparse Architecture Divergence (SAD) of N:M blocks during training N:M sparse ResNet-50 on ImageNet. The majority of blocks remain under minimal mask variation, yet a minority experience frequent mask oscillations.

we first calculate the loss-aware metric $\Delta\mathcal{L}(\mathbf{W}_{k,:})$ of all weights in an N:M block using Eq. (4) and Eq. (5). Then, we adapt the mask of weights as follows:

$$\bar{\mathbf{B}}_{k,m} = \begin{cases} 0, & \text{if } \Delta\mathcal{L}(\mathbf{W}_{k,j}) < \text{Top}(\Delta\mathcal{L}(\mathbf{W}_{k,:}), \text{M - N}), \\ 1, & \text{otherwise}, \end{cases}$$
(6)

where $m = 1, 2, ..., M$ and $\bar{\mathbf{B}}$ is the updated mask. Such mask adaptation perceptively prune-and-revive weights by looking at the effects imparted on the loss, conducting an inclusive ranking within each N:M block. Paradoxically, preceding arts that mandates the pruning of lowest magnitude weights while refurbishing those with highest gradients (Evci et al., 2020; Zhang et al., 2023a), although justified when executed across the entire weight matrix, may potentially be harmful for N:M sparsity with limited amount of weights in each block. To explain, the increment to the loss prompted by restored weights could indeed be considerably less than the disturbance to the loss distribution induced by pruned weights. Hence, our proposed LMA effectively realizes loss-aware optimization of sparse typologies.

It is also noteworthy that LMA necessitates the computation of dense gradients only intermittently, every $\Delta T$ iterations, while primarily conducting truly sparse training and weight updates at other times. This starkly contrasts previous N:M sparse training methods (Zhou et al., 2021; Fang et al., 2022; Zhang et al., 2023b), which obligate the calculation and storage of full gradients at each stage of optimization, culminating in a significantly greater training impedance relative to LMA.

### 3.3. Oscillation-aware Block Freezing

Other than LMA which enables efficient mask adaption inner each N:M block, we further stabilize the N:M sparse training process by oscillation-aware block freezing (OBF). The impetus behind OBF stems from our observations of the high-frequency mask fluctuation for each block across different LMA cycles. In particular, we employ the Sparse Architecture Divergence (SAD) (Zhou et al., 2021) to cal-

---

**Algorithm 1:** BAME for N:M Sparse Training.

**Require :** Weights $\mathbf{W}$; Initial and final iterations for mask adaption $t_i$ and $t_f$; Update interval $\Delta\text{T}$.

**Output :** Sparse weights $\bar{\mathbf{W}}$

1 **for** $t \in [t_i, \ldots, t_f]$ **do**
2    **if** $t \% \Delta T == 0$ **then**
3      Calculate $\Delta\mathcal{L}(\mathbf{W})$ via Eq. (4) and Eq. (5)
4      Obtain the adapted mask $\bar{\mathbf{B}}$ via Eq. (6)
       `// Loss-aware mask adaption`
5      Get the restricted mask $\mathbf{B}$ via Eq. (9)
       `// Oscillation-aware`
       `freezing`
6    **end**
7    $\bar{\mathbf{W}} = \mathbf{W} \odot \mathbf{B}$ `// Apply N:M mask`
8    Sparse Forward and backward propagation
9 **end**

---

culate mask fluctuation at $c$-th and $c+1$-th LMA cycle as follows:

$$SAD(\mathbf{B}_k^{c-1}, \mathbf{B}_k^c) = \sum_{m=1}^{M} |\mathbf{B}_{k,m}^{c-1} - \mathbf{B}_{k,m}^c|,$$
(7)

where $B_k^c$ denote the $k$-th block's mask at $c$-th LMA cycle. In Figure 2, we show the accumulated SAD score of different N:M blocks during N:M sparse training. The observation reveals a significantly higher frequency of mask alterations occurring in a certain number of blocks compared to others. On reflecting upon the primary intent of LMA, the apex aim during the LMA process constitutes the pruning of weights of lesser magnitude, making way for the revival of a more significant one, thereby pinpointing an enhanced position while ensuring consistent training thereafter. Nevertheless, some blocks endure recurrent deviations, alternately zeroing the weights, unquestionably inducing oscillations in loss, and thereby impeding network training. Consequently, we harness the capabilities of Exponential Moving Average (EMA) to accumulate the episodes of mask perturbations across each block, electing those exhibiting lesser fluctuations for mask evolution, thereby ensuring sta-

Table 1: Results for sparsifying ResNet-32 and MobileNet-V2 on CIFAR-10.

| Model | Method | N:M Pattern | Top-1 Accuracy (%) | Epochs (Train) | FLOPs (Train) | FLOPs (Test) |
|---|---|---|---|---|---|---|
| ResNet-32 | Baseline | - | 94.52 | 300 | 1×(3.2e16) | 1×(1.3e9) |
| ResNet-32 | ASP | 2:4 | 94.68 | 600 | 1.5× | 0.51× |
| ResNet-32 | SR-STE | 2:4 | 94.52 | 300 | 0.83× | 0.51× |
| ResNet-32 | LBC | 2:4 | 94.81 | 300 | 0.72× | 0.51× |
| ResNet-32 | **BAME(ours)** | 2:4 | **94.99** | 300 | **0.63×** | 0.51× |
| ResNet-32 | SR-STE | 1:4 | 94.52 | 300 | 0.74× | 0.26× |
| ResNet-32 | Bi-Mask | 1:4 | 94.43 | 300 | 0.49× | 0.26× |
| ResNet-32 | **BAME(ours)** | 1:4 | **94.71** | 300 | **0.39×** | 0.26× |
| ResNet-32 | SR-STE | 1:16 | 92.92 | 300 | 0.67× | 0.11× |
| ResNet-32 | Bi-Mask | 1:16 | 92.77 | 300 | 0.37× | 0.11× |
| ResNet-32 | **BAME(ours)** | 1:16 | **93.15** | 300 | **0.29×** | 0.11× |
| MobileNet-V2 | Baseline | - | 94.55 | 300 | 1×(1.4e17) | 1×(4.8e7) |
| MobileNet-V2 | SR-STE | 1:16 | 93.14 | 300 | 0.67× | 0.11× |
| MobileNet-V2 | Bi-Mask | 1:16 | 92.48 | 300 | 0.37× | 0.11× |
| MobileNet-V2 | **BAME(ours)** | 1:16 | **93.32** | 300 | **0.29×** | 0.11× |

bilized optimization for the N:M sparse network during the course of training. Concretely, we devise a vector $\mathbf{O} \in \mathbb{R}^K$, equivalent in magnitude to the count of blocks, devised for logging the frequency of mask alterations, as

$$\mathbf{O}_k^c = \gamma \, \mathbf{O}_k^{c-1} + (1-\gamma) \, SAD(\mathbf{B}_k^{c-1}, \mathbf{B}_k^c), \quad (8)$$

where $\gamma$ is the momentum of EMA updating. Then, we restrict a $\beta$ proportion of N:M blocks with the highest oscillation frequency from being updated by LMA as:

$$\mathbf{B}_{k,m}^c = \begin{cases} \mathbf{B}_{k,m}^{c-1}, & \text{if } \mathbf{O}_k^c > \text{Top}(\mathbf{O}^c, \lfloor \beta \cdot k \rfloor), \\ \bar{\mathbf{B}}_{k,m}^c, & \text{otherwise.} \end{cases} \quad (9)$$

Furthermore, within the mask adaptation selection of LMA, the occurrence of gradients is sporadic, implying the prospect of a particularly extraordinary gradient for a specified weight at a given stage. This could conceivably prompt an incorrect pruning of a substantial weight, thereby precipitating a noteworthy effect on network performance. Consequently, we confine LMA to transpire solely within weight blocks of lesser magnitudes to circumvent such inadvertent erroneous mask adaptations.

$$\mathbf{B}_{k,m}^c = \begin{cases} \mathbf{B}_{k,m}^{c-1}, & \text{if } \mathbf{O}_k^c > \text{Top}(\mathbf{O}^c, \lfloor \beta \cdot k \rfloor) \\ & \text{and } ||\mathbf{W}_k^c||_2 > \text{Top}(\hat{\mathbf{W}}^c, \lfloor \alpha \cdot k \rfloor), \\ \bar{\mathbf{B}}_{k,m}^c, & \text{otherwise,} \end{cases} \quad (10)$$

where $\hat{\mathbf{W}}_k^c = ||\mathbf{W}_k^c||_2, k = 1, 2, ..., K$. For the implementation of BAME, we follow (Jayakumar et al., 2020) to perform a three-step sparse training pipline, with $T_i$ and $T_f$ evenly divides the training schedule. In particular, we first employ gradual pruning (Zhu & Gupta, 2017) to set the

non-zeros parameters budget linearly decreased from M to the targeted N of each block in the early $T_i$ iterations. Then, we perform BAME to find the best N:M mask from $T_i$ to $T_f$. At last, we set the masks all freeze and conduct static training for the N:M sparse network within the remained iterations. The workflow for performing BAME for N:M sparse training is outlined in Alg. 1. It should be noted that although BAME effectively reduces the overhead of N:M sparse training, the sparse weights during backpropagation may not maintain N:M sparsity due to the transpose operation, which introduces specific hardware implementation challenges. To address this issue, we observe that BAME can be effectively integrated with bidirectional N:M masks (Zhang et al., 2023b) to ensure N:M sparsity throughout backpropagation; *i.e.*, we impose N:M sparsity on the backward masks of BAME based on the magnitudes of the corresponding weights. We experimentally elaborate this in the subsequent sections.

## 4. Experiment

### 4.1. Experimental Settings

**Datasets and Networks.** We validate the effectiveness of BAME by using it to train N:M sparse networks on image classification tasks on the CIFAR-10 (Krizhevsky et al., 2009) and ImageNet-1K datasets (Deng et al., 2009). For the networks, we sparsify ResNet-32 (He et al., 2016), MobileNet-V2 on CIFAR-10 dataset, and ResNet-18 (He et al., 2016), ResNet-50 (He et al., 2016), DeiT-small on ImageNet-1K dataset.

**Implementation Details.** We train N:M sparse networks

Table 2: Results for sparsifying ResNet-50 and DeiT-small on ImageNet. † means we apply the bi-directional masks (Zhang et al., 2023b) for backward propagation of BAME.

| Model | Method | N:M Pattern | Top-1 Accuracy (%) | Epochs (Train) | FLOPs (Train) | FLOPs (Test) |
|---|---|---|---|---|---|---|
| ResNet-50 | Baseline | - | 77.1 | 120 | 1×(3.2e18) | 1×(8.2e9) |
| ResNet-50 | ASP | 2:4 | 76.8 | 200 | 1.24× | 0.51× |
| ResNet-50 | SR-STE | 2:4 | 77.0 | 120 | 0.83× | 0.51× |
| ResNet-50 | LBC | 2:4 | 77.2 | 120 | 0.72× | 0.51× |
| ResNet-32 | Bi-Mask | 2:4 | 77.4 | 120 | 0.66× | 0.51× |
| ResNet-32 | MaxQ | 2:4 | **77.6** | 120 | 0.91× | 0.51× |
| ResNet-50 | BAME(ours)† | 2:4 | 77.4 | 120 | **0.59×** | 0.51× |
| ResNet-50 | **BAME(ours)** | 2:4 | 77.4 | 120 | 0.63× | 0.51× |
| ResNet-50 | SR-STE | 1:4 | 75.3 | 120 | 0.74× | 0.26× |
| ResNet-50 | Bi-Mask | 1:4 | 75.6 | 120 | 0.49× | 0.26× |
| ResNet-50 | **BAME(ours)** | 1:4 | **76.1** | 120 | **0.39×** | 0.26× |
| ResNet-50 | SR-STE | 1:16 | 71.5 | 120 | 0.69× | 0.11× |
| ResNet-50 | Bi-Mask | 1:16 | 71.5 | 120 | 0.37× | 0.11× |
| ResNet-50 | **BAME(ours)** | 1:16 | **72.0** | 120 | **0.29×** | 0.11× |
| DeiT-small | Baseline | - | 79.8 | 300 | 1×(8.9e18) | 1x(9.2e9) |
| DeiT-small | SR-STE | 2:4 | 79.6 | 300 | 0.83× | 0.11× |
| DeiT-small | Bi-Mask | 2:4 | 79.4 | 300 | 0.72× | 0.11× |
| DeiT-small | **BAME(ours)** | 2:4 | **79.7** | 300 | **0.63×** | 0.11× |

from scratch via the Stochastic Gradient Descent (SGD) optimizer, paired with a momentum of 0.9 and a batch size of 256. The initial learning rate is set to 0.1 and gradually decayed based on the cosine annealing scheduler. Following previous works, we train all networks for 300 epochs on CIFAR-10, with a weight decay of 0.005. On ImageNet, 120 epochs are given for ResNet and 300 epochs for DeiT-small. For the implementation of BAME, we set the LMA update interval $\Delta T = 100$ and 0.5 for both $\alpha$ and $\beta$ in OBF. ALL experiments are implemented based on PyTorch and executed on NVIDIA Tesla A100 GPUs.

**Performance Metrics and Baselines.** We juxtapose BAME with several state-of-the-art N:M sparsity methods, including ASP (Nvidia, 2020), SR-STE (Zhou et al., 2021), LBC (Zhang et al., 2022), Bi-Mask (Zhang et al., 2023b). We experiment with a wide range of N:M patterns for comparison, including 2:4, 1:4, and 1:16. We report the Top-1 accuracy, the training/inference float-point operations (FLOPs) and parameter burden of N:M sparse networks.

### 4.2. Image Classification

**CIFAR-10.** We first evaluate the efficacy of BAME for training sparse ResNet-32 and MobileNet-V2 on the CIFAR-10 dataset, which includes 50,000 training images and 10,000 validation images within 10 classes. Tab 1 showcases the performance comparison under different N:M patterns. BAME achieves state-of-the-art accuracy at all sce-

narios, even utilizing far fewer training FLOPs and parameters compared with other methods. For instance, BAME achieves 94.71% top-1 accuracy when training 1:4 sparse ResNet-32, surpassing the recent baseline Bi-Mask that also pursues efficient backward propagation by 0.28%. Moreover, even compared with SR-STE which conducts dense gradient calculation training, BAME still achieves better performance retention for all N:M patterns even with sparse backward propagation. For example, when training 1:16 sparse MobileNet-V2, BAME yields 93.32 top-1 accuracy, surpassing SR-STE by 0.18% while only using 0.29% training FLOPs (0.67% for SR-STE).

**ImageNet.** For the large-scale ImageNet-1K dataset that contains over 1.2 million images for training and 50,000 images for validation in 1,000 categories, we first present the quantitative results for training sparse ResNet with depths of 18 and 50, along with DeiT-small in Table 2. Again, BAME substantially enlarges the performance of existing methods, with the minimum training FLOPs by efficient weight pruning and growing. For instance, it surpasses SR-STE by 0.5% Top-1 accuracy when training 1:4 sparse ResNet-50 (76.1% for BAME and 75.3% for SR-STE), while consumes far fewer training FLOPs (0.39× for BAME and 0.74× for SR-STE). Although BAME's performance is slightly inferior to MaxQ by 0.2 Top-1 accuracy, it achieves a more than 1.54-fold reduction in training overhead. Moreover, when introducing sparsity constraints into backpropagation, BAME† still maintains notable performance advantages

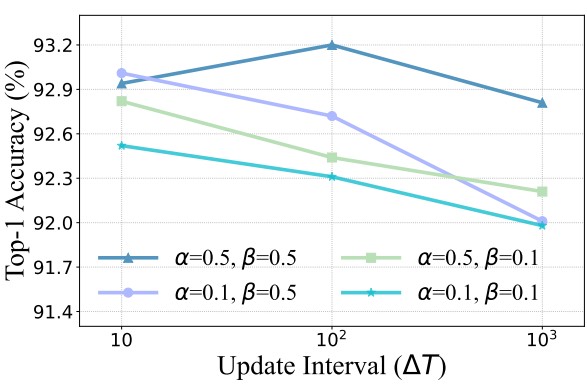

Figure 3: Ablation studies on the update schedule.

Table 3: Ablation studies on the training schedule.

| $t_i$ | $t_f$ | Top-1 Accuracy (%) | FLOPs (Train) |
|---|---|---|---|
| 0 | 100 | 91.87 | 0.07× |
| 0 | 200 | 92.57 | 0.07× |
| 0 | 300 | 92.22 | 0.07× |
| 100 | 300 | 92.81 | 0.29× |
| 200 | 300 | 93.05 | 0.41× |
| 100 | 200 | 93.15 | 0.29× |

Table 4: Ablation study on LMA and OBF.

| Method | Top-1 Accuracy (%) |
|---|---|
| Stastic | 90.03 |
| RigL | 92.01 |
| LMA | 92.98 |
| LMA+OBF | 93.15 |

compared with Bi-Mask, while readily leveraging N:M sparse tensor cores during training. It is also worth mentioning that BAME holds its advantages when training sparse DeiT-small compared with other methods, demonstrating its scalability for other types of model structures beyond convolution neural networks.

### 4.3. Performance Analysis

In this section, we provide the performance analysis of BAME, with all experiments conducted on training 1:16 ResNet-32 on CIFAR-10.

**Hyper-parameters.** We first investigate the influence of hyper-parameters within BAME, including the two restriction factors $\alpha$ and $\beta$, and the updating interval $\Delta T$. As shown in Fig.3, the best performance is obtained with $\Delta T = 100, \alpha = 0.5, \beta = 0.5$. To analyze, smaller $\alpha$ and $\beta$, larger $\epsilon$ all lead to an insufficient procedure for mask exploration during the training schedule. Setting these hyper-parameters in the contrast direction, also resulted in poor performance, which fits into our claim that high frequency of mask oscillations can unavoidably harm the training stability and lead to sub-optimal results. Nevertheless, it serves as a promising directions to automatically perform N:M sparse training without hyper-parameter choosen.

**Training Schedule.** Further, we analyze the training schedule of BAME, *i.e,* $t_i$ and $t_f$ for stooping the gradual pruning and performing mask adaption. Tab. 3 delineates the quantitative results. Intuitively, establishing a larger $t_i$ indicates an increase in training iterations for pre-training with a gradual attainment of the desired sparsity level. Though this consequently induces a significant training cost, neglecting gradual pruning simultaneously results in a considerable performance reduction. To explain, the randomly-initialized weights require a certain degree of pre-training to initiate an effective importance selection, which is validated in traditional sparsity work (Liu et al., 2021; Jayakumar et al., 2020). Regarding the mask adaptation schedule, prematurely halting BAME leads to a performance downturn due to inadequate identification of the optimal mask. In stark

contrast, prolonging BAME until the termination of training, that is, designating $t_f$ to the final iteration, results in an even more precipitous performance degradation. To explain, the freshly grown weights, initialized to zeros as per Alg. 1, mandate substantial training following restoration to enhance the performance of the sparse network.

**Mask Adaption.** At last, we investigate the effectiveness of our mask adaption methods including LMA and OBF. We set static training as the baseline, which means the binary masks are randomly initialized and kept frozen during the entire sparse training procedure. In addition, we run RigL (Evci et al., 2020), a representative method for tradition network sparse training that take weight magnitude and gradient for pruning and reviving, respectively. As shown in Tab. 4, both LMA and OBF contributes to the overall or sparse training performance.

## 5. Conclusion

N:M sparsity has become an increasingly crucial DNN compression tool, delivering functional speed ups by imposing a maximum of N non-zero constituents within M consecutive weights. We introduce BAME, a method that enhances the efficiency of the contemporary N:M sparsity methods while preserving the model's performance. BAME's fundamental principle involves carrying out loss-aware mask adaptation to prune and revitalize weights within specific N:M blocks, whilst maintaining the stability of frequently-oscillating blocks. BAME surpasses existing methods in sparsifying mainstream networks across various vision tasks, all while greatly reducing the training FLOPs and the parameter strain by keeping both sparse forward and backward propagation through training. Hopefully, BAME will not only provide practitioners with a robust N:M sparse training instrument, but also set the groundwork for further investigations into efficient N:M sparsity.

## Acknowledgements

This work was supported by the National Science Fund for Distinguished Young Scholars (No.62025603), the National Natural Science Foundation of China (No. 624B2119, No. U21B2037, No. U22B2051, No. U23A20383, No. 62176222, No. 62176223, No. 62176226, No. 62072386, No. 62072387, No. 62072389, No. 62002305 and No. 62272401), and the Natural Science Foundation of Fujian Province of China (No. 2021J06003, No.2022J06001).

## Impact Statement

This paper presents work whose goal is to advance the field of Machine Learning. There are many potential societal consequences of our work, none which we feel must be specifically highlighted here.

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
