# OpenReview forum: "BAME: Block-Aware Mask Evolution for Efficient N:M Sparse Training"
_ICML.cc/2025/Conference — ICML 2025 poster_

### Official Review · Reviewer_hLEc · 2025-03-10

**Overall Recommendation:** 4

**Summary:**

This paper introduces a novel method, BAME, which maintains consistent sparsity throughout the N:M sparse training process. Specifically, BAME ensures sparse forward and backward propagation while iteratively performing Loss-aware Mask Adaptation (LMA) and Oscillation-aware Block Freezing (OBF) to adapt the mask. This approach not only optimizes the N:M mask but also ensures training efficiency. The authors have conducted extensive experiments to validate the effectiveness of BAME.

## update after rebuttal

I think my concerns are addressed and I am happy to keep my initial positive rating.

**Claims And Evidence:**

Yes, the authors provide clear experiment results to support the theoretical foundations of LMA and OBF in BAME.

**Essential References Not Discussed:**

From the best of my knowledge, there is no essential reference that needs further discussion.

**Experimental Designs Or Analyses:**

The overall experimental design and analyses are well-structured and reasonable.

**Methods And Evaluation Criteria:**

1. The proposed method effectively addresses the current inefficiency of N:M sparsity methods.
2. The evaluation criteria align well with the image classification benchmarks, including CIFAR-10 and ImageNet.

**Other Comments Or Suggestions:**

The authors only conducted experiments on image classification tasks. It would be beneficial to perform additional experiments on tasks such as object detection to validate the effectiveness of BAME in other domains.

**Other Strengths And Weaknesses:**

Strengths:
1. The motivation behind BAME is to maintain consistent sparsity during both forward and backward propagation is very attractive. This contrasts sharply with existing methods that rely on dense backward propagation for weight updates, which can be computationally expensive. This is a necessary prerequisite for efficient N:M training on future work.
2. Extensive empirical evidence supports the effectiveness of the BAME method, particularly in keeping state-of-the-art performance while drastically reducing the training FLOPs.
3. The mathematical analysis behind Loss-aware Mask Adaptation is solid. The entire paper is well-written and easy to understand.

Weaknesses:
1. The proposed metric in Loss-aware Mask Adaptation is somewhat common in traditional sparse methods, which limits the novelty of the paper to some extent. Nevertheless, it still make clear contribution to the N:M sparsity literature.
2. The analysis of SAD is too coarse-grained in the method part. The authors did not explore the mask variations across different layers in detail, which could further optimize the OBF mechanism.

**Questions For Authors:**

As previously mentioned, does the variation in masks differ across different layers of the network? If so, could designing different OBF parameters for each layer based on these variations lead to better performance?

**Relation To Broader Scientific Literature:**

This paper effectively sets the stage for the problem of N:M sparsity and the efficiency limitations of current approaches, leading logically into the proposed solution with BAME.

**Theoretical Claims:**

I have reviewed all the theoretical analyses and found them to be sound and correct.

---

> ### Author Rebuttal · Authors · 2025-03-28
>
> We sincerely appreciate your careful review and constructive comments. Please kindly see our responses to your questions below.
>
> **Q1**: The proposed metric in Loss-aware Mask Adaptation is somewhat common in traditional sparse methods, which limits the novelty of the paper to some extent.
>
> **A1**: Thanks for this very insightful comment and we would like to share some explanations here.  While we acknowledge that pruning-and-reviving is indeed a well-established technique in sparse training literature, BAME introduces two key innovations that differentiate it from prior work: (1) block-wise loss-aware mask adaptation (LMA) for N:M sparsity, and (2) oscillation-aware block freezing (OBF) to stabilize frequently-oscillating N:M blocks. Together, these novel components enable BAME to significantly reduce training overhead while achieving state-of-the-art performance in N:M sparse network training.  We appreciate your comments and hope the above discussion clears up any misconceptions regarding our contribution in introducing BAME.
>
> **Q2**: The analysis of SAD is too coarse-grained in the method part.
>
> **A2**:  We sincerely appreciate this insightful suggestion. Following your valuable comment, we conducted additional analysis by tracking the average SAD variation per layer throughout training, with the results for sparsifying ResNet-50 at 1:16 pattern listed below (due to length limit, we random select some layers to show).
>
> | Layer | 1    | 7    | 18   | 22   | 39   | 42   | 45   | 46   |
> | ----- | ---- | ---- | ---- | ---- | ---- | ---- | ---- | ---- |
> | SAD   | 23.1 | 22.9 | 18.5 | 15.3 | 12.5 | 5.7  | 3.8  | 3.9  |
>
> The above results reveal that deeper layers indeed exhibit significantly higher SAD values compared to shallower ones. Inspired by your suggestion, we further implemented a preliminary improvement: we linearly scaled the OBF parameter from 0 to 0.5 across network layers (shallow to deep). This modification demonstrates promising results on ImageNet with ResNet-50 under 1:16 sparsity, as shown in the following table:
>
> | Method        | N:M Pattern | Top-1 Accuarcy |
> | ------------- | ----------- | -------------- |
> | OBF           | 1:16        | 72.0        |
> | Layerwise OBF | 1:16        | 72.3    |
>
> While such a heuristic modification shows potential, a more adaptive and elegant parameter allocation scheme could yield further performance gains, which we leave as a promising future work. We sincerely appreciate your expert suggestion in highlighting this valuable research avenue.
>
> **Q3**: It would be beneficial to perform additional experiments on tasks such as object detection to validate the effectiveness of BAME in other domains.
>
> **A3**: Following your professional suggestion, we further exploit the generalization ability of BAME on the object detection and instance segmentation tasks of the COCO benchmark[2]. The results are delineated as follows:
>
> - Results on object detection tasks with Faster-RCNN[3] as the backbone.
>
> | Model  | Method   | N:M  | mAP  |
> | ------ | -------- | ---- | ---- |
> | F-RCNN | Baseline | -    | 37.4 |
> | F-RCNN | SR-STE   | 2:4 | 38.2 |
> | F-RCNN | BAME     | 2:4 | 38.5 |
>
> - Results on instance segmentation tasks with Mask-RCNN[4] as the backbone.
>
> | Model  | Method   | N:M  | Box mAP | Mask mAP |
> | ------ | -------- | ---- | ------- | -------- |
> | M-RCNN | Baseline | -    | 38.2    | 34.7     |
> | M-RCNN | SR-STE   | 2:4 | 39      | 35.3     |
> | M-RCNN | BAME     | 2:4 | 39.2    | 35.4     |
>
> We will incorporate these results into our final version. Thanks for the valuable suggestion again.
>
> [1] Microsoft coco: Common objects in context. In ECCV, 2024.
>
> [2] Faster r-cnn: Towards real-time object detection with region proposal networks. In NeurIPs, 2015.
>
> [3] Mask r-cnn. In ICCV, 2017.
>
>  Your time and effort in reviewing our paper are genuinely appreciated. If there are any additional questions or points that require clarification, we would be more than delighted to engage in further discussions.

---

### Official Review · Reviewer_gJ4C · 2025-03-13

**Overall Recommendation:** 4

**Summary:**

This paper presents a novel approach for preserving sparsity in DNNs during training, with a focus on N:M sparsity. The authors introduce BAME (Block-Aware Mask Evolution), a technique that ensures both forward and backward propagation remain sparse while iteratively pruning and regrowing weights within predefined blocks. Unlike traditional methods that rely on dense backward propagation—often computationally costly—BAME offers a more efficient alternative. Experimental results on CIFAR and ImageNet show that BAME achieves performance comparable to or better than state-of-the-art methods while significantly reducing training FLOPs.

## update after rebuttal

My concerns are addressed. I will keep my rating.

**Claims And Evidence:**

The claims are well-supported by both theoretical analysis and empirical results. The authors provide a detailed theoretical proof for the efficacy of LMA, along with extensive experiment to demonstrate the performance of BAME compared with other methods.

**Essential References Not Discussed:**

No, the paper adequately covers the relevant literature and provides sufficient comparisons with other methods.

**Experimental Designs Or Analyses:**

The experimental designs are sound and well-executed. The authors conduct extensive experiments across multiple datasets (CIFAR, ImageNet) and network architectures to validate the effectiveness of BAME.

**Methods And Evaluation Criteria:**

Yes, the proposed BAME method is well-suited for the problem of N:M sparse training in DNNs. The use of block-level mask evolution is innovative and addresses the efficiency limitations of traditional methods. The evaluation criteria, including benchmark datasets like CIFAR and ImageNet, are appropriate and widely accepted in the field, ensuring the results are meaningful and comparable to prior work.

**Other Comments Or Suggestions:**

No

**Other Strengths And Weaknesses:**

*Strengths:**

1. The paper is well-structured and clearly written, making it easy to follow.
2. **Innovative Methodology**: The paper introduces a fresh perspective on N:M sparse training by focusing on block-level sparsity evolution. This approach effectively balances efficiency and performance, showcasing significant potential for reducing the computational costs associated with dense training.
3. **Theoretical Rigor**: The theoretical proofs are thorough and provide a solid foundation for understanding the design principles behind the LMA metric. Additionally, the visualization of mask oscillations effectively validates the design rationale of OBF.
4. **Empirical Evidence**: The experimental results are extensive and demonstrate the efficacy of BAME across various network architectures and datasets. The results show that BAME can achieve substantial reductions in training FLOPs without compromising model accuracy.

**Weaknesses:**

1. While BAME demonstrates lower training overhead compared to MaxQ, its performance appears to be slightly inferior to MaxQ in certain cases.
2. Another minor limitation of the proposed method is its reliance on hyperparameters. Future work could explore ways to automate N:M sparse training to reduce the need for such specific hyperparameter tuning.

**Questions For Authors:**

I keep up with the literature in this area.

**Relation To Broader Scientific Literature:**

The key contributions of this paper build on prior work in N:M sparse training. Specifically, the authors address the efficiency limitations of existing methods that rely on dense backward propagation, which is computationally expensive. By introducing block-level mask evolution, BAME offers a more efficient alternative that aligns with recent trends in reducing the computational cost of training DNNs. The paper effectively situates itself within the broader literature by comparing BAME to state-of-the-art methods and demonstrating its advantages.

**Theoretical Claims:**

Yes, the theoretical claims regarding the LMA method appear to be correct. The proofs are well-structured and logically sound. No issues were identified in the theoretical analysis.

---

> ### Author Rebuttal · Authors · 2025-03-28
>
> We sincerely appreciate your positive and motivating comments. Please kindly see our response to your comment below.
>
> **Q1**: While BAME demonstrates lower training overhead compared to MaxQ, its performance appears to be slightly inferior to MaxQ in certain cases.
>
> **A1**:  We appreciate this insightful comment. We acknowledge that our method may exhibit marginally inferior performance to MaxQ in some scenarios. However, we would like to highlight that our approach achieves a significant reduction in training FLOPs—from 0.91× to 0.59× for training 2:4 sparse ResNet-50 compared to dense training. Thus, while maintaining comparable performance to MaxQ, our method offers substantial advantages in training efficiency.
>
> **Q2**: Another minor limitation of the proposed method is its reliance on hyperparameters.
>
> **A2**:  We appreciate this constructive feedback and acknowledge that BAME does involve several hyperparameters. However, we would like to clarify that most of these hyperparameters follow established conventions from prior sparse training methods [1,2]. As such, they are considered well-established defaults in the field and typically do not require extensive tuning—for instance, the update interval ΔT is fixed at 100, following standard practice. Despite this, we recognize the potential effectiveness of automatically performing N:M sparse training without choosing hyperparameters, and we earmark this aspect for future work.
>
> [1] Rigging the lottery: Making all tickets winners. In ICML, 2020
>
> [2] Sparse Training via Boosting Pruning Plasticity with Neuroregeneration. In NeurIPs, 2021.
>
> We sincerely appreciate the time and diligence you’ve taken to participate in the review of our paper. If you have further questions, we are more than glad to discuss with you.

---

> > ### Comment · Reviewer_gJ4C · 2025-04-03
> >
> > Thanks for the rebuttal. My concerns are addressed.

---

> > > ### Author Response · Authors · 2025-04-03
> > >
> > > We sincerely appreciate your strong support and great interest in our work. We are also delighted that our rebuttal has effectively addressed your questions.

---

### Official Review · Reviewer_2Nc7 · 2025-03-23

**Overall Recommendation:** 3

**Summary:**

The paper introduces a novel approach called BAME (Block-Aware Mask Evolution) for training N:M sparse networks in an efficient manner. The authors argue that prior works often rely on dense gradient updates, which leads to considerable overhead. Instead, BAME keeps the network consistently N:M sparse throughout both the forward and backward passes. The core idea comprises two components: Loss-Aware Mask Adaption (LMA): and Oscillation-aware Block Freezing (OBF). Experimental results on CIFAR-10 and ImageNet show that BAME achieves accuracy comparable to or surpassing prior N:M sparsity methods, while consuming significantly fewer training FLOPs

**Claims And Evidence:**

The paper’s main claim is that BAME’s consistent N:M sparsity (applied throughout forward and backward passes) can significantly reduce training overhead without sacrificing accuracy. The authors present empirical results on CIFAR-10 and ImageNet using models such as ResNet, MobileNet, and DeiT, demonstrating that BAME achieves on-par or better accuracy compared to prior N:M sparsity methods at substantially lower training FLOPs.

One limitation, however, is that these claims rely on theoretical FLOPs counts. The authors do not provide actual training or inference speed measurements, such as latency, throughput, or wall-clock time, which would give more direct evidence of real-world performance gains.

**Essential References Not Discussed:**

I did not identify any critical missing references that would drastically affect the paper’s context.

**Experimental Designs Or Analyses:**

- The experiments are well-structured: multiple N:M patterns (2:4, 1:4, 1:16) are tested on CIFAR-10 and ImageNet with widely used architectures. The ablation studies around hyperparameters (α,β,ΔT) and scheduling (when to start and stop mask adaption) support the validity of the method.

- The results consistently show superior or on-par accuracy with lower training FLOPs. The authors also compare with relevant baselines (SR-STE, ASP, LBC, Bi-Mask, MaxQ).

- The experiments seem sound overall, but additional real-world speedup measurements (wall-clock time) on hardware supporting N:M sparsity (e.g., A100 GPU) might further demonstrate the actual training speed benefits.

**Methods And Evaluation Criteria:**

The proposed method applies periodic “pruning-and-regrowth” within each N:M block, guided by LMA and OBF that freezes blocks prone to mask instability.

- Top-1 Accuracy

- Training/Test FLOPs

- NM spasrity

While these metrics are typical and demonstrate clear benefits (especially in theoretical FLOPs), the paper omits direct measurements of actual training/inference speed (e.g., wall-clock time, throughput, latency on real hardware). Including such real-world measurements would strengthen the argument that the reduced FLOPs indeed translate to practical speedups and resource savings.

**Other Comments Or Suggestions:**

I would recommend clarifying the initialization and final freeze phases in even more detail: how exactly are the pruned weights re-initialized? Are they set to zero or restored from historical values, etc.?

**Other Strengths And Weaknesses:**

Strengths:

- Presents a well-motivated approach for block-level mask evolution.

- Simple idea but demonstrates strong empirical results across multiple networks/datasets.

- Reduces training FLOPs while retaining or improving accuracy compared to prior methods.

Weaknesses:


- More real-world training speed tests (beyond theoretical FLOPs) would be helpful to confirm the practical efficiency on hardware that supports N:M acceleration.

- The choice and tuning of certain hyperparameters (α,β,ΔT) might require domain knowledge, although the paper provides some guidance through ablation.

**Questions For Authors:**

- For newly “revived” weights, are they always re-initialized to zero or do they retain their old (pre-pruned) value [1]? If always zero-initialized, does that hamper potential recovery for large updates? If not, how do you track historical states?

-  Have you measured actual training speed on an NVIDIA A100 or similar GPU to confirm that the 0.29× or 0.39× training FLOPs factor translates into a similar wall-clock reduction? If so, please provide more context.

- You mention that OBF (Oscillation-aware Block Freezing) identifies blocks with high-frequency mask changes and freezes them to improve training stability. However, one might wonder if these frequent updates are actually a signal that those blocks are “sensitive” or “important.” In other words, rather than straightforwardly freezing such blocks, could there be a more nuanced approach—one that better leverages this apparent sensitivity while still mitigating the risk of oscillation?


[1] CHEX: CHannel EXploration for CNN Model Compression. CVPR 2022

**Relation To Broader Scientific Literature:**

- BAME aligns with the growing literature on sparse training (RigL, SET, SNIP) and N:M sparsity (ASP, SR-STE, LBC, Bi-Mask).

- The core novelty is combining a loss-aware local pruning/regrowth scheme with an oscillation detection approach specifically tailored for N:M blocks.

**Theoretical Claims:**

The submission does not present extensive theoretical proofs beyond first-order Taylor approximations for analyzing loss impact.

---

> ### Author Rebuttal · Authors · 2025-03-28
>
> We sincerely appreciate your positive and motivating comments. Please kindly see our response to your concerns below.
>
> **Q1**:  Actual training or inference speed measurement of BAME.
>
> **A1**:  We appreciate this constructive feedback. We first clarify that **reporting N:M sparsity patterns and theoretical FLOPs (rather than empirical acceleration ratios) is a standard practice in the N:M sparsity community** [1–3]. This is because N:M sparsity is inherently a hardware-software co-design problem: algorithmic works in this domain typically focus on ensuring that weight matrices satisfy N:M sparsity constraints during forward/backward propagation, while retaining the performance. Critically, once N:M sparsity is achieved in either phase (forward or backward), the acceleration effect becomes deterministic given certain hardware support[4,5]. For instance, [5] reports a 2.17× speedup for 2:8 sparse training when both forward and backward passes are sparse (equivalent to our BAME, T-mask[2], Bi-mask[3]), while methods with dense backward passes (e.g., SR-STE[1]) achieve only 1.33× speedup. Currently, the only open-source hardware framework is NVIDIA Ampere Sparse Tensor Core[4] that support 2:4 inference. Below are our test inference latency results on an NVIDIA A100 GPU (batch size=512):
>
> | Method | N:M | Latency per Batch (ms) | Top-1 Acc.|
> | - | - | -| - |
> | ResNet-50 |- | 59.88|77.4|
> | BAME | 2:4 | 37.19|77.4|
>
> This demonstrates practical inference acceleration (1.61×) for BAME. Regarding training latency, existing N:M training frameworks are neither open-source nor easily deployable without specialized hardware expertise. Due to limited resources, **we prioritized algorithmic innovation (aligning with the community norms) to report only theortical FLOPs**. While FLOPs remain a standard metric for cross-method comparison, we will add the result of inference latency and clarify this point in our final version to avoid any misunderstanding.
>
> **Q2**: Theoretical proofs beyond first-order Taylor approximations for analyzing loss impact.
>
> **A2**: We fully agree that more rigorous theoretical derivations exist—such as second-order Hessian-based analysis. However, we employ first-order approximation due to its efficiency, as the derived metric w*g_w is readily available during training. This ensures training efficiency, and first-order approximations have also been widely validated as effective in reflecting loss impact[8], though exploring beyond them remains a promising future direction. Thanks very much for this insight.
>
> **Q3**: Hyperparameters might require domain knowledge.
>
> **A3**: We acknowledge that BAME does involve several hyperparameters, yet most follow established sparse training recipes[8] and require minimal tuning. Despite this, we recognize the potential of automatically performing N:M training without choosing hyperparameters, and we earmark this aspect for future work.
>
> **Q4**:  Clarify the initialization and final freeze phases in even more detail.
>
> **A4**: We re-initialize the pruned weights to their historical values rather than set them to zero. This is directly motivated by Equation 6 of our manuscript, which evaluates the impact of restoring a pruned weight on the loss by leveraging its historical state. For the storage of historical states, we directly store and freeze the values of pruned weight. We ensure that the above clarification will be added to our final version.
>
> **Q5**: The relationship between frequent updates and block importance.
>
> **A5**: Thank you for sharing this insightful comment. Indeed, mask oscillation has been a persistent research point in the community. Zhou et al. [1] showed mask fluctuations correlate with loss reduction, where excessive variations yield suboptimal results. Their SR-STE solution increases weight decay on pruned weights to stabilize masks. The point you raised presents an intriguing research direction, as both our OBF and SR-STE seem to be heuristic to avoid mask oscillation. Perhaps these sensitive blocks may require specialized evaluation metrics to assess weight significance that avoids fluctuation without compromising performance. Addressing this challenge demands substantial algorithmic innovation, which we reserve for future work.
>
> We sincerely appreciate the time and effort you have dedicated to reviewing our paper. Should you have any further inquiries, please let us know and we would be more than delighted to engage in further discussion with you.
>
> [1] Learning N: M fine-grained structured sparse neural networks from scratch. In ICLR, 2021
>
> [2] A Provable and Efficient Method to Find N:M Transposable Masks. In NeurIPs, 2021
>
> [3] Bi-directional Masks for Efficient N:M Sparse Training. In ICML, 2022.
>
> [4] Nvidia a100 tensor core gpu architecture, 2020
>
> [5] Efficient N:M Sparse DNN Training Using Algorithm, Architecture, and Dataflow Co-Design. In IEEE TCAD, 2023.
>
> [6] Rigging the lottery: Making all tickets winners. In ICML, 2020.

---

### Decision · Program_Chairs · 2025-05-01

**Decision:**

Accept (poster)

**Comment:**

The paper introduces a novel approach called BAME (Block-Aware Mask Evolution) for training N:M sparse networks aiming to preserve the sparsity through both forward and backward propagation. The core BAME novelty is combining a loss-aware local pruning/regrowth scheme with an oscillation detection approach specifically tailored for N:M blocks. All reviews are very positive, recognize the novelty of the findings, and appreciate the presentation of the paper. Therefore, the AC suggests acceptance and encourages the authors to include the rebuttal discussion in the final version of the paper.